# Various Anti-HSPA2 Antibodies Yield Different Results in Studies on Cancer-Related Functions of Heat Shock Protein A2

**DOI:** 10.3390/ijms21124296

**Published:** 2020-06-16

**Authors:** Dorota Scieglinska, Damian Robert Sojka, Agnieszka Gogler-Pigłowska, Vira Chumak, Zdzisław Krawczyk

**Affiliations:** Center for Translational Research and Molecular Biology of Cancer, Maria Sklodowska-Curie National Research Institute of Oncology Gliwice Branch, 44-101 Gliwice, Poland; damian.sojka@io.gliwice.pl (D.R.S.); agnieszka.gogler-piglowska@io.gliwice.pl (A.G.-P.); vira.chumak@io.gliwice.pl (V.C.); zdzislaw.krawczyk@io.gliwice.pl (Z.K.)

**Keywords:** HSPA2, heat shock protein family A, cancer biomarker, immunohistochemistry, antibody specificity, proteotoxic stress

## Abstract

Heat shock proteins (HSPs) constitute a major part of the molecular chaperone system and play a fundamental role in cell proteostasis. The HSPA (HSP70) family groups twelve highly homologous HSPA proteins. Certain HSPAs are regarded as important cancer-related proteins, prospective therapeutic targets for cancer treatment, and also as potential cancer biomarkers. Heat Shock Protein A2 (HSPA2), a testis-enriched chaperone and one of the least characterized members of the HSPA family, has recently emerged as an important cancer-relevant protein with potential biomarker significance. Nevertheless, conflicting conclusions have been recently drawn both according to HSPA2 role in cancer cells, as well as to its prognostic value. In this work we have shown that one of the serious limitations in HSPA2 protein research is cross-reactivity of antibodies marketed as specific for HSPA2 with one or more other HSPA(s). Among non-specific antibodies were also those recently used for HSPA2 detection in functional and biomarker studies. We showed how using non-specific antibodies can generate misleading conclusions on HSPA2 expression in non-stressed cancer cells and tumors, as well as in cancer cells exposed to proteotoxic stress. Our findings addressed concerns on some published studies dealing with HSPA2 as a cancer-related protein.

## 1. Introduction

Heat Shock Proteins (HSPs) constitute a major part of the molecular chaperone system and play a fundamental role in cell proteostasis. HSPs, on the basis of molecular weight are classified into six major families [1]. Among them is the HSPA (previous name HSP70) family that groups twelve HSPA proteins which share an exceptionally high amino-acid sequence homology.

Numerous experimental investigations revealed that certain members of the HSPA family are important cancer-related proteins and prospective therapeutic targets for cancer treatment [2,3]. HSPAs are also regarded as potential biomarkers related, e.g., with cancer stage, grade and prognosis [4,5,6]. One of the family members, recently suggested as an essential contributor to malignant phenotype is HSPA2, a protein originally recognized as testis-enriched and with a proven critical role in male fertility [7]. Although research on the involvement of HSPA2 in cancer is in early stage, results of several clinically oriented correlation studies revealed association between HSPA2 expression and patients survival (Table 1). However, conflicting conclusions on HSPA2 prognostic value were achieved in studies employing classical immunohistochemistry (IHC) method, as compared to studies analyzing mRNA expression. For example, in breast cancer, an IHC-based study showed that a high level of HSPA2 in primary tumors is related with unfavorable clinicopathological characteristics [8], and shorter overall and disease-free survival [9]; while analysis of the transcriptomic data revealed that high levels of HSPA2 mRNA predict a favorable outcome [6,10,11]. Intriguingly, such a discrepancy was not detected in the studies on non-small cell lung carcinoma (NSCLC) and pancreatic adenocarcinoma. In both these cases, high levels of HSPA2 mRNA and protein expression were associated with a worse prognosis [6,12,13,14].

Ambiguities observed in breast cancer, may result either from the fact that HSPA2 expression is regulated at posttranscriptional level or, alternatively, from some technical limitations in HSPA2 detection. Indeed, specific immunodetection of each particular HSPA protein is challenging due to potential cross-reactivity of the antibody with other HSPA isoform(s). HSPA2 shares 86.3% amino-acid sequence homology with HSPA8 (HSC70), a constitutively expressed housekeeping chaperone; 83.5% with HSPA1 (aliases: HSP70-1, HSP70i, HSP72, HSP70.1), a stress-inducible protein which is frequently overexpressed in cancer cells; and 78.3% with HSPA6 (HSP70B’), a strictly stress-inducible, short-lived chaperone that is absent in cells under physiological conditions [7]. HSPA2 shares a lower, although still meaningful, similarity to HSPA5 (GRP78) and HSPA9 (GRP75) (60.9% and 46%, respectively). Specific detection of HSPA proteins in tumors is even more complex due to cancer heterogeneity and conditions of tumor microenvironment which means that different sets of HSPAs can be concomitantly overexpressed in respective cancer cells. Unfortunately, the results of cross-reactivity testing are not provided by manufacturers of the commercial anti-HSPA2 antibodies. In our experience, only the non-commercial, monospecific rabbit polyclonal anti-HSPA2 antibody has been confirmed to specifically bound to its intended antigen in NSCLC model [12,15].

Bearing in mind that the usage of non-validated antibodies is a serious limitation to credibility of immunodetection results, in this study we compared the specificity of six selected commercial anti-HSPA2 antibodies, mostly the products used for HSPA2 detection in cancer cells and tissues in previously published studies (Table 1). Antibodies were tested for potential cross-reactivity with recombinant HSPA1, HSPA6 and HSPA8. The specificity of antibodies was also examined in a model system comprising genetically engineered isogenic cell lines differing by HSPA2 expression status, as well as in cancer cells under proteotoxic stress conditions in vitro, and in primary lung tumors. Our results indicate that specific immunodetection of HSPA2 is not assured by the majority of antibodies tested. Thus, some published data on HSPA2 expression and function in human cancers should be considered with caution and may need to be reexamined.

## 2. Results

### 2.1. Reactivity of Commercial Anti-HSPA2 Antibodies with Highly Homologous HSPA Proteins

We tested the cross-reactivity of six selected commercial anti-HSPA2 antibodies, of which four were used in previously published studies (Table 1). Technical details (vendor, clonality, immunogen and dilution used for detection) are collected in the Material and Methods section. Antibodies were probed against purified recombinant HSPA2-GST (fusion protein), HSPA1, HSPA6 and HSPA8. Our results indicate that five anti-HSPA2 antibodies react additionally with one or more other HSPA(s) (cross-reactivity). Only Abcam antibody binds selectively and exclusively to HSPA2-GST (Figure 1). A summary of these results is shown in Table 2.

### 2.2. Detection of the Endogenous HSPA2 in Cells Cultured under Non-Stressful Conditions

Recombinant protein may exhibit key structural differences in comparison to native one. Therefore, we tested whether commercial anti-HSPA2 antibodies ensure detection of native HSPA2 in NCI-H1299 cells, both wild-type and genetically engineered, according to HSPA1 and HSPA2 expression. We used a previously constructed NCI-H1299 cell line variant—sh-A2.4, with shRNA-mediated stable HSPA2 knockdown. We have previously confirmed that HSPA2 is expressed at highly reduced levels in the sh-A2.4 cell line [18]. Commercial antibodies were also tested in sh-A1.S cells with confirmed stable shRNA-mediated knockdown of HSPA1 expression [18]. As controls, we used NCI-H1299 cells stably transduced with vector-encoding non-targeting shRNA (sh-luc) and wild-type NCI-H1299 cells (wt).

Abcam and Proteintech antibodies detected highly decreased levels of HSPA2 in HSPA2-deficient sh-A2.4 cells (Figure 2A), while the Novus antibody indicated only slight decrease of HSPA2 level in these cells. Surprisingly, the Sigma antibody detected lower levels of HSPA2 in HSPA1-deficient (sh-A1.S) cells than in HSPA2-deficient (sh-A2.4) cells (Figure 2A). In case of other anti-HSPA2 antibodies, we did not observe any effect of HSPA1 knockdown on their performance (Figure 2A).

Next, we tested suitability of anti-HSPA2 antibodies to detect HSPA2 in cells producing increased levels of either HSPA2 or HSPA1. We used a pLVX-A2 cell line producing high level of HSPA2 (CMV promoter-directed overproduction) (Figure 2B) and cells expressing increased levels of HSPA1 (pCMV-A1; transient transfection with vector encoding HSPA1 under the CMV promoter) (Figure 2C). As a control, we used pLVX cells, stably transduced with the ‘empty’ pLVX vector (Figure 2B) or cells transiently transfected with the ‘empty’ pCMV vector (pCMV) (Figure 2C), respectively.

Five out of six tested antibodies were able to detect high accumulation of HSPA2 protein in HSPA2-overexpressing cells (pLVX-A2). Instead, the Sigma antibody showed only a moderate increase in antigen concentration in HSPA2-overproducing cells (Figure 2B).

Accumulation of HSPA1 influenced results of HSPA2 detection when using Proteintech and Sigma antibodies (Figure 2C): both reagents generated higher immunosignal in HSPA1-overproducing than in control cells. Conversely, HSPA2 detection performed using Abcam or Novus antibody generated a lower-intensity signal in HSPA1-overproducing than in control cells (Figure 2C). These observations confirmed that antibodies from Sigma and Proteintech cross-react with HSPA1 in cells producing this antigen at high levels.

### 2.3. Detection of the Endogenous HSPA2 in Cells under Proteotoxic Stress Conditions

A massive overproduction of stress-inducible HSPAs (mainly HSPA1 and HSPA6) is stimulated by proteotoxic stress, a condition caused, among others, by inhibited or impaired proteasome function. To validate the reliability of HSPA2 detection in cellular environment highly enriched in stress-inducible HSPAs, we evoked proteotoxic stress in MCF7 cells using MG132, a reversible proteasome inhibitor. We used MCF7 cells, since a massive accumulation of HSPA2 following MG132 treatment can be detected in these cells, as shown recently using the Proteintech antibody [9]. For detection of stress-inducible HSPA1 and HSPA6, and constitutive HSPA8, we used commercial antibodies (Table 3). Anti-HSPA6 and anti-HSPA8 antibodies were specific for their intended antigen (Figure 3A, Table 2), while two anti-HSPA1 antibodies used (Enzo, Life Sciences (SPA810); Proteintech; Table 2) reacted with both HSPA1 and HSPA6 (Figure 3A). First, we confirmed that MG132 treatment increased the levels of inducible HSPA1 and HSPA6 but had no impact on HSPA8 expression (Figure 3B). HSPA6 expression was highly induced in cells exposed to high doses (10–20 µM) of proteasome inhibitor (Figure 3B).

Figure 3C shows that depending on anti-HSPA2 antibody selection one can observe highly variable changes in HSPA2 expression after MG132 treatment. All antibodies that cross-reacted with HSPA1 (Santa Cruz Biotechnology) or with HSPA1 and HSPA6 (Proteintech, Sigma) showed a massive increase in signal intensity after proteasome inhibition (Figure 3C). Conversely, no increase in signal intensity was observed when Abcam or Novus antibodies were used. The first one showed even reduced immunostaining in response to MG132 (Figure 3C). Given that Abcam antibody, but not Novus antibody, shows no cross reactivity with other HSPAs (Figure 1), this result shows that HSPA2 expression decreases following proteasome inhibition in MCF7 cells.

### 2.4. The Impact of Anti-HSPA2 Antibody Used for IHC Detection on Conclusions about HSPA2 Expression in Tumors

IHC assessment of an antigen expression is a frequently used approach in biomarker studies. In this aspect, the usage of validated primary antibody is an absolute must to obtain reliable results. The IHC technique was used in several studies aimed at evaluating HSPA2 as potential cancer biomarker (details in Table 1) [9,12,14,16,17]. However, in only one of these studies [12] was an anti-HSPA2 antibody with previously proven specificity used [15].

Here, we compared results of HSPA2 detection by IHC using four commercial anti-HSPA2 antibodies. We analyzed the same NSCLC samples, in which the status of HSPA2 expression was previously determined with an antigen-affinity purified custom monospecific rabbit polyclonal antibody [12]. Figure 4 shows that only antibody from Abcam recapitulated previously established pattern of HSPA2 expression in NSCLC samples. In both cases, IHC staining was present only in some NSCLC samples and exclusively in cytoplasm and/or nuclei of cancer cells (Figure 4, Appendix A). Other antibodies stained both cancer and stromal cells, and staining was present also in samples previously characterized as HSPA2-negative (Figure 4, Appendix A).

## 3. Discussion

Members of the HSPA family, due to the extremely high homology of amino acid sequence, are a good example of technically challenging antigens. HSPA groups more than 10 highly homologous proteins, of which some have been intensively studied for many years, while others remain poorly characterized. One of the latter is HSPA2, a chaperone protein initially considered as testis-specific, but now known to be also expressed in a cell type-specific manner in selected somatic organs. Recently, HSPA2 has drawn wider attention as a possible cancer-related protein and potential cancer biomarker. In the presented study, we showed that a large part of the current belief on cancer-related HSPA2 features stems from research performed with the use of non-validated and/or non-specific antibodies. We are not referring to whether the conclusions drawn from some of the cited studies are correct or not. In this study, we rather aimed at evaluating the reliability of current data on the role of HSPA2 in the cancer process, indicating that some of the results obtained using anti-HSPA2 antibodies should be revised.

We showed that only one from among six commercial anti-HSPA2 antibodies ensured specific detection of HSPA2 in cancer cells and tissues. This antibody marketed by Abcam showed no cross-reactivity with constitutively expressed HSPA8, stress-inducible HSPA1 (usually expressed at low levels even in the absence of stress) and HSPA6, a strictly stress-inducible protein (absent under physiological conditions). We have used this antibody in our recent studies on HSPA2 expression and function in epidermal keratinocytes and NSCLC cells [18,19]; it was also used in one study aimed at evaluating potential value of HSPA2 as a prognostic biomarker in pancreatic adenocarcinoma [14]. Importantly, consistent conclusions on HSPA2 as a marker of poor outcome for pancreatic cancer patients were obtained analyzing HSPA2 expression in primary tumors at mRNA level [13]. It has to be underlined that this antibody is a good equivalent to our custom-made monospecific polyclonal anti-HSPA2 antiserum, which was used by us to detect HSPA2 in the testis, keratinocytes and some other normal somatic cells, as well as in tumors [7,15,20].

Except the product marketed by Abcam, other commercial anti-HSPA2 antibodies were non-specific, therefore they do not meet quality criteria referred for IHC-based biomarker study. Antibodies marketed by Santa Cruz Biotechnology, Proteintech and Sigma cross-react with HSPA1, HSPA6 and HSPA8. In an animal study, the Proteintech antibody was used for HSPA2 detection in mouse testis [21], while in humans it was recently used to search for prognostic significance of HSPA2 in breast cancer [9]. The latter study showed that an elevated HSPA2 expression correlated with shorter overall and disease-free survival [9]. This finding stands in striking contrast with results of other studies which show that high *HSPA2* mRNA expression can predict a favorable outcome [6,10,11].

Monospecific rabbit polyclonal anti-HSPA2 antibody marketed by Sigma-Aldrich company belongs to Prestige Antibodies collection developed and validated within the Human Protein Atlas project. According to our knowledge, the antibody was used to study the role of HSPA2 in rodent testes [22,23,24]. HSPA2 is abundantly expressed in developing spermatogenic cells, in which the stress-inducible HSPA1 and HSPA6 are not expressed, therefore HSPA2 seems to be the only specific antigen present in the testis. However, the use of this antibody for detection of HSPA2 in extra-testicular cells and tissues that contain other HSPAs is not recommended. Additionally, the validation of other commercial antibodies used for HSPA2 detection in the testes (e.g., antibody SAB1405970 from Sigma-Aldrich [25,26]; antibody sc-100760 from Santa Cruz Biotechnology [27]) should precede their usage for HSPA2 analysis in somatic tissues or tumor samples.

The anti-HSPA2 antibody marketed by Santa Cruz Biotechnology was used in studies on prognostic value of HSPA2 in hepatocellular carcinoma [17] and esophageal SSC [16]. However, in both papers a reference to antibody is too limited to recognize the identity of product used (no information about catalog number, host or clonality; Table 1). Of note, currently, Santa Cruz Biotechnology stopped marketing any anti-HSPA2 antibodies. A total lack of reference for antibody source also pertains to recent study on HSPA2 in NSCLC [28]. Consequently, an attempt to reproduce the results of the abovementioned studies is a ‘mission impossible’.

In this work, we also observed inconsistent results when testing the specificity of mouse monoclonal antibodies distributed by different vendors, e.g., R&D System and Novus Biologicals, which were considered identical since they share the same clonal origin and catalogue number. As we found both products cross-reacted with HSPA8, but surprisingly R&D products bound more strongly to HSPA8 than to HSPA2. This observation shows the need for the validation of products having exactly the same labels but being distributed by different vendors.

The use of non-specific antibodies can explain controversial results obtained in studies on HSPA2 in cells undergoing proteotoxic stress due to proteasome inhibition. Here, using a highly specific antibody, we found that the level of HSPA2 decreased in MCF7 cells exposed to proteasome inhibitor. In contrast, a massive accumulation of HSPA2 in MCF7 cells after proteasome inhibition was detected by others with the use of Proteintech antibody [9]. Thus, our finding showing that Proteintech antibody cross reacts with HSPA1 and HSPA6 resolves the abovementioned discrepancy. At the same time, this result rises the necessity to verify the mechanisms regulating HSPA2 stability via involvement of ubiquitin ligase RNF144A, as proposed by Yang et al., [9].

It seems that due to a high homology of HSPA proteins, a generation of specific anti-HSPA antibody is highly dependent on selection of an appropriate immunogen. In our opinion there is only a marginal chance (if any) that the whole-length HSPA2 protein, as an immunogen, will enable a specific antibody to rise. The HSPA2-specific Abcam antibody was generated using a recombinant fragment corresponding to human HSPA2 aa 450–650 (C-terminal). Several functional studies on HSPA2 in cancer cells (details on these studies are collected in Appendix A) were performed using custom-made rabbit anty-HSPA2 polyclonal antisera raised against the short peptide corresponding to murine HSPA2 aa 611–628 (C terminal), a fragment showing the lowest homology to other HSPAs sequences. Such an antiserum was first produced by Rosario et al. [29] for detection of murine HSPA2; later, a similar methodology was used for the detection of human HSPA2 by Scieglinska et al., [12,15,30] and Garg et al. [31]. The purification protocol, e.g., affinity purification using antigen derived from human testis [15] versus protein G chromatography [32] was the main difference comparing methods of antiserum production for HSPA2 detection in human. Noteworthy, only the specificity of antiserum developed by Scieglinska et al. [15] was confirmed. 

The data presented in this paper clearly show that using non-validated and/or non-specific antibodies may undermine the reliability of conclusions on HSPA2 as a prognostic tumor marker. The general aim of our research is to raise researchers’ awareness that the antibodies sold as specific for a particular HSPA protein may also recognize other HSPA(s). In our opinion, a demonstration that anti-HSPA antibody selectively recognizes a protein of a 70-kDa weight is not sufficient proof of its specificity. Although this problem is recognized by experts in the field, less experienced researchers may be not aware of such a technical trap which, in fact, can severely influence the conclusions drawn from their studies. Therefore, the correct validation process of commercial and custom-made anti-HSPA antibodies should obligatorily include testing for potential cross-reactivity with other HSPAs. The data documenting this step should be an integral part of a manuscript. Unquestionably, well-characterized, highly specific antibodies are essential tools for basic research and in clinical diagnostic. Our duty, as researchers, is to create a research culture taking care of technical perfection in order to generate reproducible results and gain good knowledge.

## 4. Materials and Methods

### 4.1. Cell Culture and Experimental Conditions

NCI-H1299 (non-small cell lung carcinoma, CRL-5803), and MCF7 cells (breast carcinoma, HTB-22) were purchased from American Type Culture Collection (Manassas, VA, USA) and cultured at 37 °C under standard conditions (5% CO_2_, 95% humidity, 21% O_2_ concentration). NCI-H1299 and MCF7 cells were grown in RPMI or DMEM/F12 media (Sigma-Aldrich, St. Louis, MO, USA), respectively supplemented with 10% heat-inactivated fetal bovine serum (EuRx, Gdańsk, Poland) and antibiotics. Cells were regularly checked for mycoplasma contamination. The protocol for the stable knockdown of HSPA2 or HSPA1 expression in NCI-H1299 by lentiviral shRNA was described in detail in [18]. MCF7 cells, 24 h after plating, were exposed to a MG132 (Selleck Chemicals, Houston, TX, USA) inhibitor for 24 h. Working solutions were prepared fresh from stock solution before each experiment in culture medium (without antibiotics).

### 4.2. Transient Transfection

NCI-H1299 cells (2 × 10^5^/well, 6-well dish) were transfected with pCMV-HSPA1 plasmid (gift from Prof. A. Żylicz/D. Walerych, pcDNA3.1 backbone [33]) or “empty” pCMV plasmid without transgene using 1.5 µg of plasmid and Lipofectamine 3000 (Life Technologies, Carlsbad, CA, USA). Further experiments were performed 48 h after transfection.

### 4.3. Generation of Lentiviral Expression Vectors and Lentiviral Transduction

Lentiviral vector encoding the HSPA2 protein under CMV promoter was constructed by insertion of the HSPA2 coding sequence from pEF1-HSPA2 plasmid [30] between EcoRI and BamHI restriction sites of the pLVX-Puro plasmid (Clontech, Takara Bio, Mountain View, CA, USA) using In-Fusion® HD EcoDry™ Cloning Plus (Clontech, Takara Bio, Mountain View, CA, USA) according to the manufacturer’s manual. Primers used in cloning were as follows: forward 5’-CTCAAGCTTCGAATTCATGTCTGCCCGTGGCCCG-3 and reverse 5’-TAGAGTCGCGGGATCCTTAATCCACTTCTTCGATGGTGGG-3’. The correctness of plasmids generation was confirmed by Sanger sequencing. Lentiviruses were produced by transfecting plasmids into HEK293T packaging cells according to the manufacturer’s instructions (Lenti-X™ Packaging Single Shots and Lenti-X shRNA Expression System, Clontech, Takara Bio, Mountain View, CA, USA). A viral titer  >  5 × 10^5^ IFU/ml in virus-containing supernatants was confirmed using Lenti-X GoStix (Clontech, Takara Bio, Mountain View, CA, USA). NCI-H1299 cells (2 × 10^5^/well, 6-well plate) were transduced at 24 h after plating with supernatants containing lentiviruses for 24 h at 37 °C with the addition of 4 μg/mL polybrene. The pLVX-A2 cell line, which overexpresses HSPA2 protein, was established by the lentiviral transfer of the pLVX-Puro-HSPA2 plasmid. The control pLVX cell line was established by transfer of the pLVX-Puro vector. Stably transduced cells which survived puromycin selection (1 μg/ml, Sigma Aldrich, St. Louis, MO, USA), were used in further experiments.

### 4.4. Protein Extraction and Western Blot Analysis

Cells were seeded in 6-cm dishes with a maximum confluency of 50–70%. To prepare total protein extracts, cells were lysed by scrapping in RIPA buffer (1× PBS, 1% NP-40, 0.1% SDS, 0.5% SDC, 50 mM NaF, 1 mM PMSF) supplemented with protease inhibitor mixture (Roche). Total protein content was determined using Protein Assay Kit (Bio-Rad, Hercules, CA, USA). Total proteins (25–35 μg) were fractionated by SDS-PAGE on 8% polyacrylamide gels and transferred on nitrocellulose membrane using Trans Blot Turbo system (Bio-Rad, Hercules, CA, USA) for 10 min. The membrane was blocked in 5% nonfat milk/TTBS (0.25 M Tris–HCl (pH 7.5), 0.15 M NaCl, and 0.1% Tween-20), and incubated with anti-HSPA primary antibody (Table 3). Actin was detected as a protein loading control using mouse monoclonal β-actin antibody (1:20000; clone AC15, A3854, RRID AB_262011, Merck KGaA, Darmstadt, Germany,). Antibody-antigen interaction was detected using HRP-conjugated secondary antibody: goat anti-mouse (1:5000) (AP124P, RRID AB_90456, Millipore Billerica, MA, USA), goat anti-rabbit (1:2000) (AP132P, RRID AB_90264, Millipore Billerica, MA, USA), chicken anti-goat (1:2000) (sc-2953, RRID AB_639230, Santa Cruz Biotechnology Inc., Dallas, TX, USA), and visualized using Clarity ECL Western Blot Substrate (Bio-Rad; Hercules, CA, USA). Detection of β-actin was used as a loading control.

Ten nanograms (10 ng) of recombinant HSPA2-GST fusion protein (H00003306-P01, Abnova, Taoyuan City, Taiwan), HSPA6 (ADI-SPP-762-E, Stressgen Biotechnologies Corporation, Victoria, BC, Canada), HSPA1 and HSPA8 (gifts from Prof. A. Żylicz) were separated by SDS-PAGE on 8% polyacrylamide gel and immobilized on nitrocellulose membrane (Sartorius Stedim Biotech GmbH, Goettingen, Germany). Immunodetection was performed as described above.

### 4.5. Immunohistochemistry (IHC)

Samples from our collection of formalin-fixed, paraffin-embedded, surgically resected NSCLC primary tumors that differ by HSPA2 expression status were used (details in [12]). The protocol of the study was approved by the Ethics review board (no KB/430-66/12; 14-Nov-2012). The IHC procedure was performed as described previously [12]. Briefly, citrate-mediated (0.01 M buffer pH 6.0), heat-induced antigen retrieval was performed prior to IHC [32]. IHC staining was performed using an IMPRESS Universal kit (Vector Laboratories Inc. Burlingame, CA, USA). The IHC controls included omission of the primary antibody.

## Figures and Tables

**Figure 1 ijms-21-04296-f001:**
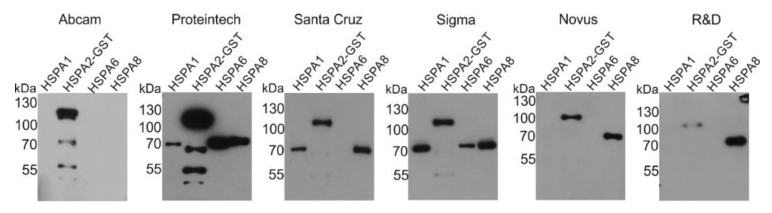
Reactivity of six commercial anti-HSPA2 antibodies to different recombinant HSPAs. Immunoblots show binding of commercial anti-HSPA2 antibodies to recombinant HSPA1 (69.92 kDa), HSPA2-GST (96.03 kDa), HSPA6 (71.028 kDa) and HSPA8 (70.898 kDa) proteins (predicted molecular weight in kDa is indicated). In total, 10 ng of recombinant protein was blotted and detected using anti-HSPA2 antibody (*n* = 2).

**Figure 2 ijms-21-04296-f002:**
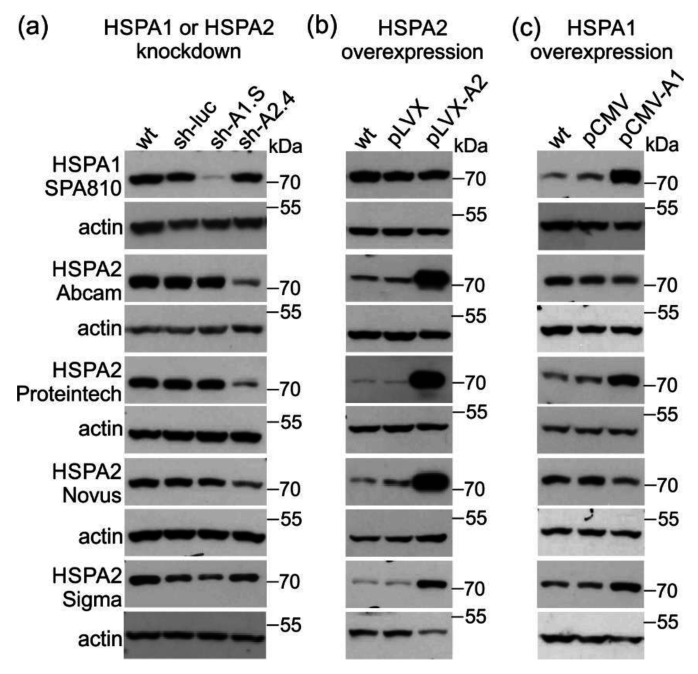
Western blot detection of endogenous HSPA2 in NCI-H1299 cells using commercial anti-HSPA2 antibodies. (**a**) Reactivity of commercial anti-HSPA2 antibodies to native HSPA2 protein in wild-type (wt) and lentivirally transduced cell sublines: sh-luc, control cells expressing non-targeting shRNA sequence; sh-A1.S, cells deficient in HSPA1 (expressing HSPA1-targeting shRNA sequence); sh-A2.4, cells expressing HSPA2-targeting sequence (shRNAs sequences were described in [18]. SPA810, antibody distributed by Enzo, Life Sciences. (**b**) HSPA2 overexpression in lentivirally modified cell variants; pLVX, control cells transduced with empty vector pLVX-Puro; pLVX-A2, cells transduced with pLVX-Puro vector bearing HSPA2 protein coding sequence. (**c**) Detection of HSPA2 in cells overproducing HSPA1 protein (pCMV1-A1) and in control cells (pCMV). Altogether, 25–35 µg of total protein extracts were blotted and immunodetected using the anti-HSPA2 antibody. Representative immunoblots are shown (*n* ≥ 3), actin—a protein loading control.

**Figure 3 ijms-21-04296-f003:**
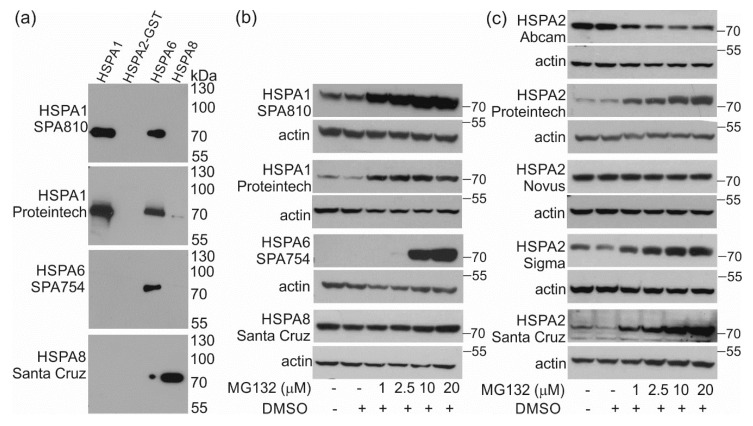
Response of HSPAs system to proteotoxic stress in MCF7 cells exposed to MG132 proteasome inhibitor. (**a**) Reactivity of antibodies used for detection of stress-inducible HSPA1 and HSPA6 and constitutive HSPA8 proteins against human recombinant HSPA1, HSPA2-GST, HSPA6 and HSPA8 proteins. SPA810 and SPA754, antibodies distributed by Enzo, Life Sciences. (**b**) Detection of endogenous HSPA1, HSPA6, HSPA8 and (**c**) HSPA2 in non-treated and MG132-treated cells. Cells were incubated with the inhibitor for 24 h, total protein extracts (30–35 µg) were used for Western blot analysis. Representative immunoblots are shown (*n* ≥ 3), actin was used as a protein loading control.

**Figure 4 ijms-21-04296-f004:**
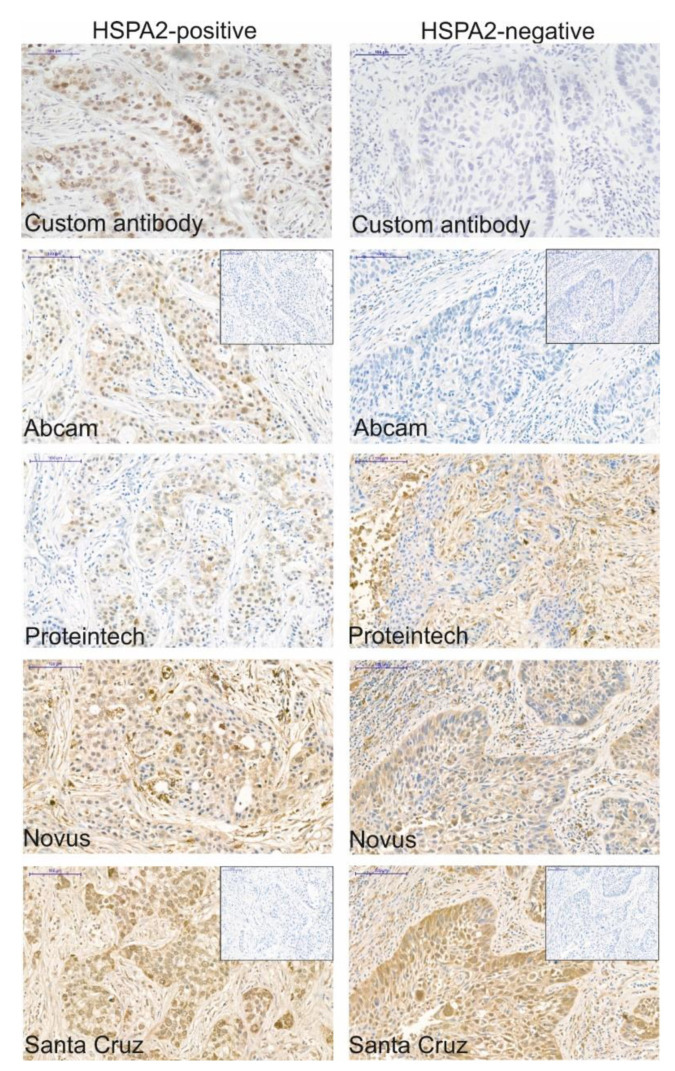
Results of immunohistochemical staining with the use of specific and validated custom anti-HSPA2 antibody [12] and commercial anti-HSPA2 antibodies in NSCLC tumor samples were incubated with primary antibody overnight at 4 °C. Two pairs of HSPA2-positive and -negative samples were analyzed; representative images are shown. The summary on antibody cross-reactivity and on technical details are collected in Table 2 and Table 3, respectively. The scale bar represents 100 µm.

**Table 1 ijms-21-04296-t001:** Correlation between patient survival and HSPA2 expression at mRNA or protein level in primary tumors.

Tumor Type	Method	Material/Group Size	Results	Ref
BRCA	Analysis of gene expression data	TCGA^1^	High expression of HSPA2 mRNA was associated with better OS	[10]
BRCA	Analysis of gene expression data	TCGA^1^/*n* = 1085	High HSPA2 mRNA expression was associated with a good prognosis	[11]
BRCA	Analysis of gene expression data	TCGA^1^/*n* = 1101 KM plotter^2^/*n* = 5143	High expression of HSPA2 mRNA was associated with better OS for patients from TCGA cohort, better OS and RFS in KM plotter cohort	[6]
BRCA	IHC/rabbit pAb 12797-1, Proteintech Group	primary tumor FFPE/*n* = 166	High expression of HSPA2 was associated with shorter OS and RFS	[9]
BRCA, ACC, KICH, UCS, H&N SCC, pancreatic AD, KIRP, UCEC, rectum AD, BLGG, THCA, HCC, prostate AD	Analysis of gene expression data	TCGA^1^	Favorable prognosis (survival Z-score < 0) associated with HSPA2 mRNA overexpression	[6]
esophageal SCC	IHC/Santa Cruz Biotechnology Inc	primary tumor FFPE/*n* = 120	High HSPA2 expression was an independent negative prognostic factor for OS and DFS	[16]
HCC	IHC/goat pAb, Santa Cruz Biotechnology Inc.	primary tumor FFPE/*n* = 119	High HSPA2 expression was an independent negative prognostic factor for OS	[17]
lung NSCLC	IHC/monospecific rabbit pAb (custom made)	primary tumor FFPE	High HSPA2 expression correlated with shorter OS in the stage I-II subgroup.	[12]
pancreatic AD	qRT-PCR	primary tumor/*n* = 104	High HSPA2 mRNA level was an independent negative prognostic factor for OS	[13]
pancreatic carcinoma	IHC/rabbit mAb EPR4596 Abcam	primary tumor FFPE/*n* = 80	High HSPA2 level was independent negative prognostic factor for RFS and OS	[14]
SKCM, AML, lung AD, BLC, lung SCC, OV, colon AD, CC, DLBC, PCPG, GBM, SARC	Analysis of gene expression data	TCGA^1^	HSPA2 overexpression associated with poor survival	[6]

^1^ TCGA database contains data on mRNA gene expression levels in primary tumor samples measured using the RNA-Seq; ^2^ KM plotter database contains gene expression data from GEO (Affymetrix microarrays only), European Genome-phenome Archive (EGA) and TCGA.

**Table 2 ijms-21-04296-t002:** Cross-reactivity of anti-HSPA antibodies.

	Catalog Number/RRID	Source	Reactivity			
			A1	A2	A6	A8
HSPA1	ADI-SPA-810-F/AB_311860	Enzo, Life Sciences	+	-	+	-
	10995-1-AP/AB_2264230	Proteintech Group	+	-	+	+/-
HSPA2	Ab108416/AB_10862351	Abcam	-	+	-	-
	12797-1-AP/AB_2119687	Proteintech Group	+	+	+	+
	sc-1600010/n.d.	Santa Cruz Biotechnology Inc.	+	+	-	+
	HPA000798/AB_1079090	Sigma-Aldrich	+	+	+	+
	MAB6010/AB_1964604	Novus Biologicals LLC	-	+	-	+
	MAB6010/AB_1964604	R&D Systems Inc.	-	+	-	+
HSPA6	ADI-SPA-754/AB_10615942	Enzo, Life Sciences	-	-	+	-
HSPA8	sc-7298/AB_627761	Santa Cruz Biotechnology Inc.	-	-	-	+

**Abbreviations:** n.d., no data; RRID, Research Resource Identifiers; +, evident; +/-, weak, -, not detected.

**Table 3 ijms-21-04296-t003:** Anti-HSPA antibodies.

	Host/Clonality	Clone	Catalog Numer/RRID	Source	Dilution WB/IHC	Immunogen
HSPA1	Mo/mAb	C92F3A-5	ADI-SPA-810-F/AB_311860	Enzo, Life Sciences, Famingdale, NY, USA	1:5000/n.d.	Native human HSPA1 protein
	Ra/pAb		10995-1-AP/AB_2264230	Proteintech Group, Rosemont, IL, USA	1:25000/n.d.	HSP70 fusion protein (Ag1446)
HSPA2	Ra/mAb	EPR4596	Ab108416/AB_10862351	Abcam, Cambridge, UK	1:5000/1:5000	Recombinant fragment matching to human HSPA2 aa 450-650
	Ra/pAb		12797-1-AP/AB_2119687	Proteintech Group, Rosemont, IL, USA	1:3000/1:50	HSPA2 Fusion Protein (Ag3539)
	G/pAb		sc-1600010 (K12)/nd	Santa Cruz Biotechnology Inc., Dallas, TX, USA	1:2000/1:200	Peptide matching to internal region of human HSPA2
	Ra/pAb		HPA000798/AB_1079090	Sigma Aldrich, St. Louis, MO, USA	1:2000/n.d.	Recombinant Protein Epitope Signature Tag (PrEST)
	Mo/mAb	520608	MAB6010/AB_1964604	Novus Biologicals LLC, CO, USA	1:5000/1:200	Recombinant human HSPA2 Lys529-Gly616
	Mo/mAb	520608	MAB6010/AB_1964604	R&D Systems Inc., Minneapolis, MN, USA	1:2000/n.d.	Recombinant human HSPA2 Lys529-Gly616
HSPA6	Mo/mAb	165f	ADI-SPA-754/AB_10615942 or AB_2039272	Enzo, Life Sciences, Famingdale, NY, USA	1:3000/n.d.	Synthetic peptide mapping near the C-terminus of human HSPA6
HSPA8	Mo/mAb	B-6	sc-7298/AB_627761	Santa Cruz Biotechnology Inc., Dallas, TX, USA	1:7500/n.d.	Epitope mapping 583-601 at the C-terminus of human HSPA8

**Abbreviations:** G, goat; IHC, immunohistochemistry; mAb, monoclonal antibody; n.d., no data; Mo, mouse; pAb, polyclonal antibody; Ra, rabbit; RRID, Research Resource Identifiers; WB, Western Blot.

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
