# Peer review of "Various Anti-HSPA2 Antibodies Yield Different Results in Studies on Cancer-Related Functions of Heat Shock Protein A2"

_ijms, 2020, doi:10.3390/ijms21124296_

Round 1

Reviewer 1 Report

In this paper, Scieglinska et al. aimed at determining the specificity of several commercial anti-HSPA2 antibodies. The concept is interesting and important as most researchers are dependent on commercial antibodies. However, as said by the authors the aim of the study is not to criticize all the studies performed with non-specific anti-HSPA2, as many of these studies used complementary methods, but to alert the scientific community and to explain discrepancies between studies on the use of HSPA2 as a cancer biomarker.

Major points:

  • In figure 1 the blot revealed with the Abcam antibody is false. Actually, the antibody only detects HSPA6.
  • In figure 1, please remind the awaited MW of different HSPs. Can you explain the different bands obtained in some cases, particularly for HSPA2-GST transfected cells? Are these bands observed with an anti-GST?
  • Line161: the authors said “we confirmed that MG132…. a significant increase in HSPA1”. However, in this experiment HSPA6 is also increased by MG132, and the authors showed in table 2 that the SPA810 antibody also recognizes HSPA6. I don’t think that the conclusion is really sure. Please shade your sentence.
  • The results in figure 3A panel with anti-HSPA1 from Proteintech is different from the result described in Table 2 (no detection of HSPA8 in fig3A). Can you explain these discrepancies?
  • I think that Figure 4 would be more convincing if the authors performed the experiments on other tumor samples. If possible a quantification will strengthen the results and conclusions.
  • In Material and methods, the sequencing of constructs coding for HSPA1 and HSPA2 must be confirmed to be sure that if an anti-HSPA recognize these proteins it is not due to a wrong construct.
  • Is there a possibility to be sure that the commercial recombinant HSPs used in this study are pure?

Minor points:

  • The reference number 8 is not mentioned in Table 1.
  • Line 59-61. Cancer heterogeneity is not the only thing that can influence HSPA expression. The tumor microenvironment (especially immune cells) can also do it. Please add this commentary.
  • Line 130-132: the authors said that the Sigma antibody showed only a moderate increase in HSPA2. However the actin staining is lower too in this well, so please shade this conclusion.
  • In figure 2A the title of this panel is HSPA2 knockdown. However the authors also showed HSPA1 knockdown cells in this panel. Please correct this.

Reviewer 2 Report

This is a well written paper that aims to show that commercial antibodies that are being sold to detect HSPA2 have very low specificity. The authors test antibodies from several vendors and in various approaches including recombinant protein detection, expression in cell lines by western blot and most importantly IHC. Their conclusions are well supported by the data shown. This is a very technical paper, but its results have clear relevance for the research lines established around HSP in cancer.

in figure 1 abcam labels are probably wrong, as it is said in the text that it recognizes HSPA2 but the figure shows supposed recognition of HSPA6. the actual bands correspond most likely to HSPA2, so the figure labels are likely not correct.

Given the fact that HSP8 is the most crossreacted Heat Shock Protein and that its expression is constitutive in most cells. Wouldn’t be a good idea to repeat experiments as in figure 2 but knockin down HSPA8 in the cells?

Is it possible that a different antigen retrieval technique allows better specificity of any of the cross reactive antibodies tested? For example could be this the case of the Sigma antibody mentioned in the discussion?
